# National trends in utilization and outcomes of elective open and minimally invasive colostomy reversal: A NSQIP analysis

Kevin Tabibian[1], Nam Yong Cho[1], Oh Jin Kwon[1], Esteban Aguayo[1], Ayesha P. Ng[1], Courtney Obasohan[1], Dariush Yalzadeh[1], Arjun Chaturvedi[1], Peyman Benharash[1,2], Hanjoo Lee[3]*

1 Center for Advanced Surgical and Interventional Technology (CASIT), David Geffen School of Medicine, University of California, Los Angeles, California, United States of America, 2 Department of Surgery, David Geffen School of Medicine at the University of California Los Angeles, Los Angeles, California, United States of America, 3 Department of Surgery, Division of Colon and Rectal Surgery, Harbor-University of California, Los Angeles Medical Center, Torrance, California, United States of America

* h.lee@dhs.lacounty.gov

## Abstract

### Background

Minimally invasive approach for reversal of Hartmann's procedure remains understudied. This study examined the outcomes associated with open and minimally invasive approaches for colostomy reversal in a national cohort.

### Methods

The 2012–2022 American College of Surgeons National Surgical Quality Improvement Program participant use file data was queried to identify all adult (≥18 years) patients undergoing elective open or minimally invasive colostomy takedown. Multivariable regression models were developed to assess the associations between operative modalities and outcomes of interest, including overall complications (cardiac, respiratory, infectious, wound, renal and thromboembolic postoperative sequelae as well as reoperation and transfusion), operative duration, postoperative length of stay, and 30-day Readmissions.

### Results

Among the 20,163 patients who underwent colostomy takedown during the study period, 6,180 (30.7%) had a minimally invasive reversal. Utilization of minimally invasive colostomy reversal increased from 18.2% in 2012 to 41.9% in 2022 (nptrend < 0.001). Following risk adjustment, minimally invasive colostomy takedown was associated with reduced odds of overall complications compared to the open approach (AOR 0.56, 95% CI 0.51–0.62). The minimally invasive approach was

**Data availability statement:** The dataset was provided by the American College of Surgeons National Surgical Quality Improvement Program (NSQIP) as the "Participant Use File." NSQIP data is available for researchers via Data Use Agreement. HIPAA-compliant, deidentified data can be requested by any researcher affiliated with NSQIP-participating hospitals at: [https://www.facs.org/quality-programs/acs-nsqip/participant-use/puf-form].

**Funding:** The author(s) received no specific funding for this work.

**Competing interests:** The authors have declared that no competing interests exist.

associated with decremental operative duration by 16.9 minutes (95% CI 13.6 to 20.2 minutes) and postoperative length of stay by 1.70 days (95% CI 1.56 to 1.84 days), as well as decreased odds of 30-day readmission (AOR 0.75, 95% CI 0.67–0.85).

## Conclusions

Over the past decade, utilization of minimally invasive colostomy reversal has more than doubled and yielded lower overall complication rates compared to the open approach. Our findings suggest that the minimally invasive approach may be appropriate for colostomy takedown in suitable cases.

## Introduction

An estimated 150,000 patients require ostomy creation as part of complex bowel resection across the United States each year [1]. While colostomy remains a lifesaving intervention when anastomosis is not feasible after resection, it poses ongoing risks of both short- and long-term complications such as stomal necrosis, sepsis and stenosis [2]. Physiology and function notwithstanding, colostomies may profoundly affect psychosocial well-being [3]. Additionally, colostomy is associated with a significant financial burden with average out-of-pocket costs ranging from $100–300 per month [4,5]. Consequently, a significant proportion of patients opt for colostomy reversal when clinically feasible, with the reversal rate estimated to be upwards of 40% [6].

Historically, the open technique has been the most common approach for Hartmann's colostomy reversal, but a recent study has reported an increasing trend in utilization of minimally invasive approaches [7]. Celentano et al. demonstrated both open and minimally invasive colostomy takedowns to be associated with comparable operative time and 30-day mortality rates [8]. In contrast, various retrospective cohort studies have demonstrated minimally invasive reversal to be associated with reduced hospital length of stay and a more rapid return of bowel function compared to the open surgery [7,9,10]. Nevertheless, the clinical advantages of minimally invasive over open colostomy takedown remain inconclusive in current literature.

In the present work, we characterized trends and clinical outcomes associated with open and minimally invasive colostomy reversal techniques in a national cohort of patients. We hypothesized the utilization of minimally invasive colostomy takedown to have increased over the study period. We additionally hypothesized the minimally invasive approach to be associated with reduced risks of clinical complications, postoperative length of stay (LOS) and 30-day readmission compared to open approach.

## Methods

This was a retrospective cohort study of the 2012–2022 American College of Surgeons National Surgical Quality Improvement Program (ACS NSQIP) participant use files. The data utilized in this study were sourced from the ACS NSQIP and its participating hospitals [11]. These entities have not verified the data and assume no

responsibility for the statistical validity of the analyses, or the conclusions drawn by the authors. Given the de-identified nature of the ACS NSQIP, this study was deemed exempt from full review by the Institutional Review Board at the University of California, Los Angeles.

All adults (≥18 years) undergoing colostomy reversal of Hartmann's procedure, were tabulated using previously cited Current Procedural Terminology (CPT) codes 44227 and 44626 [7,12]. The identified cases were subsequently stratified into minimally invasive (laparoscopic and robotic-assisted) and open cohorts. Records with emergent operation, non-elective admission, or missing key data (age, sex, American Society of Anesthesiologists (ASA) classification, or functional status), were excluded from further analysis (Fig 1). Ileostomy status was identified using relevant International Classification of Diseases, 9th and 10th Revisions (ICD-9/10) diagnosis codes: ICD-9 (V44.2 and V55.2) and ICD-10 (K94.10, K94.11, K94.12, K94.13, K94.19, Z43.2, and Z93.2) [13,14]. Patient records involving a postoperative diagnosis code pertaining to ileostomy care were excluded from the study (14.0%). Demographic and clinical variables were defined using ACS NSQIP-provided data elements and included age, sex, body mass index (BMI), diabetes, hypertension, smoking status, chronic obstructive pulmonary disease (COPD), chronic steroid use, weight loss, functional status, and ASA Classification [11].

Notably, in 2022, two additional variables—*robot_used* and *unplanned_conv_open*—were introduced into the ACS NSQIP dataset, enabling a robust subgroup analysis. Specifically, *robot_used* was utilized to distinguish robotic-assisted from laparoscopic approaches within the minimally invasive cohort, and *unplanned_conv_open* was used to exclude unplanned conversions to open surgery.

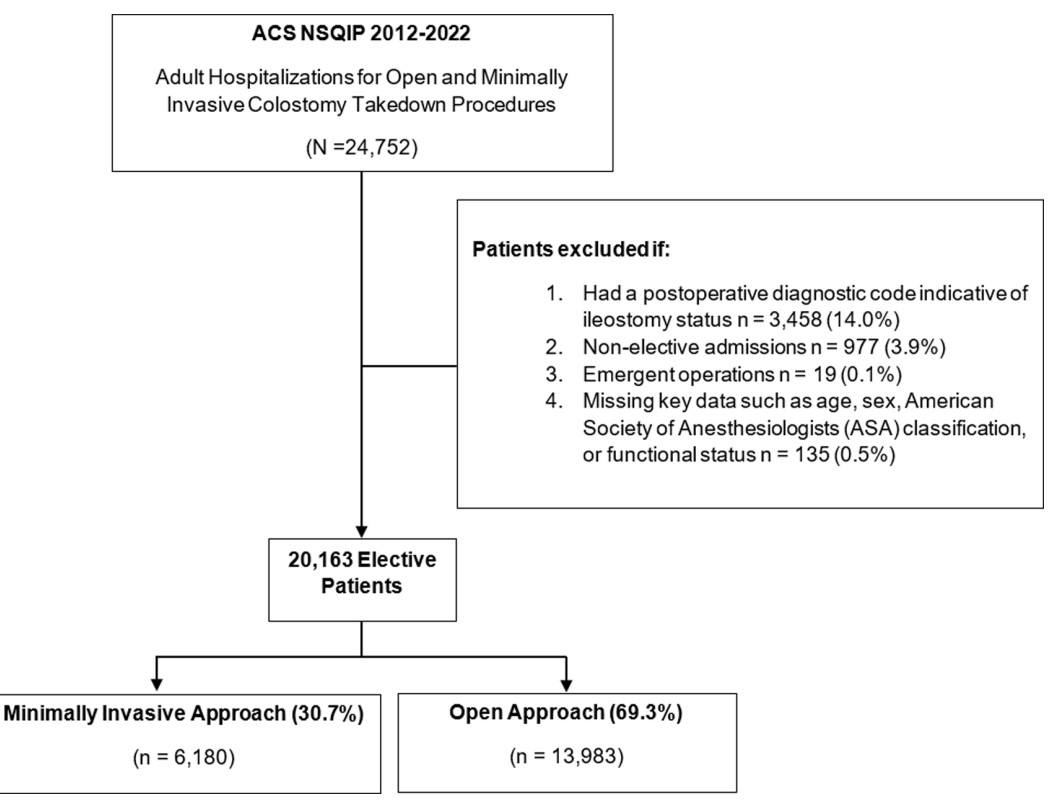

**Fig 1. Flow chart of study cohort undergoing open or minimally invasive colostomy reversal from 2012–2022.** A subgroup analysis was conducted on patients within the 2022 cohort to individually assess the laparoscopic and robotic-assisted approaches. *ASA*, American Society of Anesthesiologists.

The primary outcome of interest was overall complications. Of note, overall complications consisted of cardiac (cardiac arrest and myocardial infarction), respiratory (pneumonia, reintubation, prolonged ventilation > 48 hours), infectious (superficial and deep wound infections, organ space infection, urinary tract infection, sepsis, septic shock), wound (dehiscence), renal (acute renal insufficiency and failure) and thromboembolic (pulmonary embolism and deep vein thrombosis) complications as well as reoperation and transfusion within 30 days of index procedure. Secondary endpoints included operative duration, LOS, and 30-day readmissions.

Categorical and continuous variables were reported as proportions (%) or medians (if skewed) with interquartile range (IQR), respectively. The Pearson's $\chi^2$ and Mann–Whitney $U$ tests were utilized for unadjusted comparisons. The temporal trends were assessed using Cuzick's nonparametric test (nptrend) [15]. Variable selection was performed using elastic net regularization, which combines the Least Absolute Shrinkage and Selection Operator (LASSO) and ridge regression penalties to reduce bias and increase generalizability [16]. Multivariable logistic and linear regressions were developed to evaluate the association between colostomy reversal approach and outcomes of interest. Marginal analysis was conducted using the Stata *margins* command, yielding point estimates of risk adjusted values and corresponding confidence intervals. Regression outputs are reported as adjusted odds ratios (AOR) or beta coefficients (*β*) with 95% confidence intervals (95% CI). Statistical significance was set at an α of 0.05. All statistical analysis was performed using Stata 16.1 (StataCorp, College Station, TX).

## Results

Of 20,163 patients meeting study criteria, 6,180 (30.7%) underwent a minimally invasive colostomy takedown while 13,983 (69.3%) had an open approach. Utilization of minimally invasive approach increased significantly over the 11-year study period, rising from 18.2% in 2012 to 41.9% in 2022 (nptrend < 0.001) (Fig 2).

Both minimally invasive and open cohorts had comparable distributions of age, sex, and BMI (Table 1). In addition, functional status did not differ significantly between study cohorts (Table 1). The minimally invasive cohort had lower rates of COPD (3.8 vs 5.1%, p < 0.001), hypertension (44.6 vs 46.8%, p = 0.01), and smoking (21.2 vs 25.3%, p < 0.001).

On unadjusted analysis, minimally invasive colostomy takedown had lower rates of 30-day overall complications (13.2 vs 23.1%, p < 0.001), compared to open (Table 2). The minimally invasive cohort had a lower incidence of respiratory (1.5 vs 2.9%, p < 0.001), infectious (8.6 vs 15.9%, p < 0.001), and wound complications (0.6 vs 1.7%, p < 0.001) (Table 2). Additionally, the minimally invasive cohort had shorter operative time (185 [133–254] vs 196 [138–270] minutes, p < 0.001) and LOS (4 [43 –5 ] vs 5 [5,4 –7 ] days, p < 0.001) compared to open cohort. Minimally invasive colostomy reversal also had a lower incidence of nonhome discharge (3.1 vs 5.4%, p < 0.001) and 30-day readmissions (7.8 vs 10.6%, p < 0.001).

Following risk adjustment, minimally invasive reversal was associated with decreased odds of overall complications compared to the open approach (AOR 0.56, 95% CI 0.51–0.62). Marginal analysis indicated a consistent reduction in the risk-adjusted rate of overall complications over time for both minimally invasive and open approaches (Fig 3). Notably, minimally invasive colostomy takedown was associated with a reduced odds of respiratory (AOR 0.61, 95% CI 0.47–0.79), infectious (AOR 0.54, 95% CI 0.48–0.61), and wound complications (AOR 0.39, 95% CI 0.26–0.57) (Table 3). With open as reference, minimally invasive takedown was associated with decremental operative duration by 16.9 minutes (95% CI 13.6 to 20.2 minutes) and LOS by 1.70 days (95% CI 1.56 to 1.84 days) as well as reduced odds of 30-day readmission (AOR 0.75, 95% CI 0.67–0.85) and nonhome discharge (AOR 0.59, 95% 0.48–0.71). (Table 3). Marginal analysis of the LOS model showed that, over the study period, minimally invasive colostomy takedown was consistently associated with a shorter predicted postoperative length of stay compared to open reversal. Furthermore, both operative modalities exhibited a decremental LOS over time (Fig 4).

In a subgroup analysis of 2,253 patients undergoing colostomy reversal in 2022, 190 (8.4%) were found to have undergone unplanned conversion to open. Following adequate exclusion, 328 (15.9%) underwent robotic-assisted procedures, and 578 (28.0%) had a laparoscopic reversal. Robotic-assisted colostomy reversal had a higher rate of overall

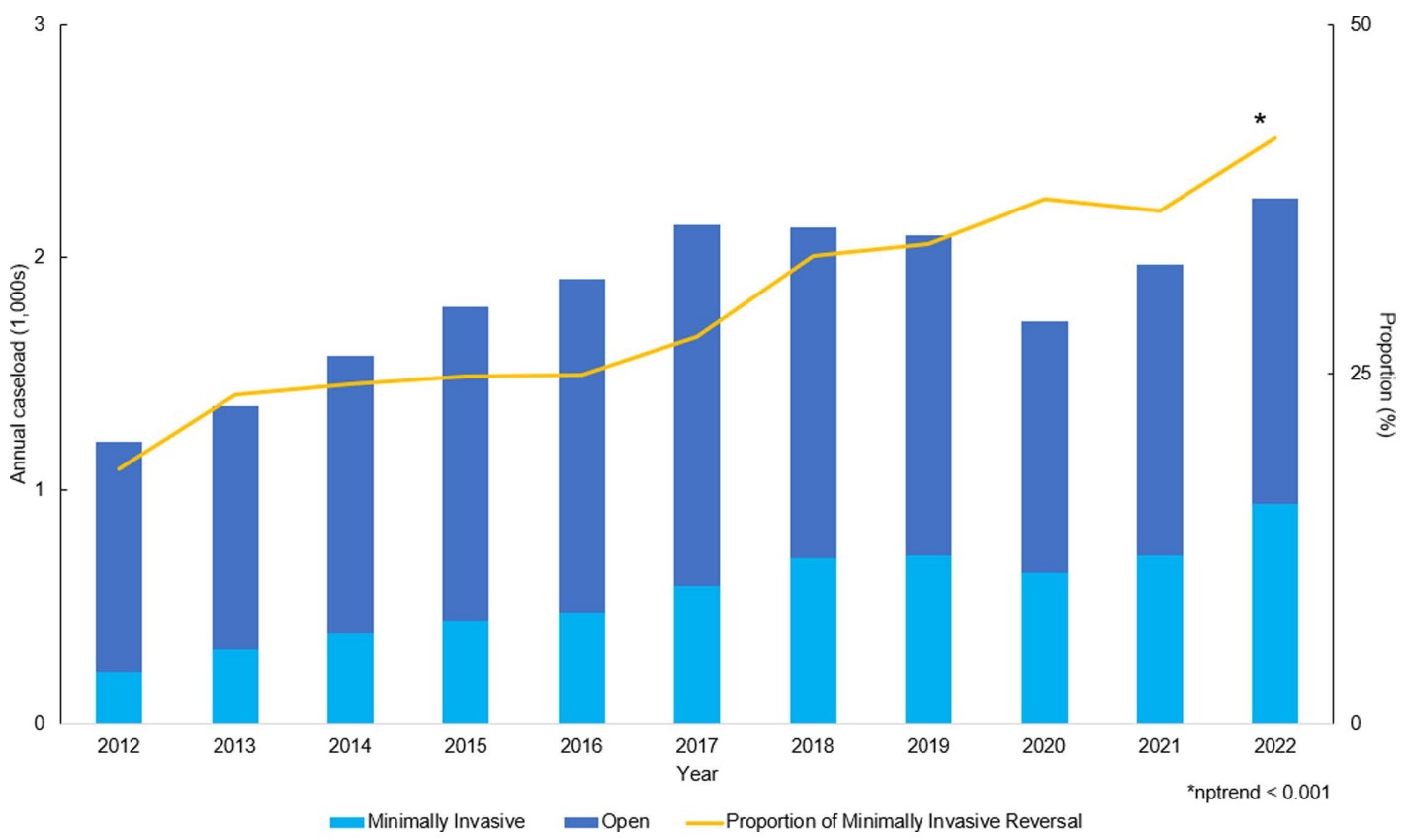

**Fig 2. Annual trends in the volumes of open and minimally invasive colostomy reversal, 2012-2022.** The proportion of minimally invasive procedures significantly increased over the study period (nptrend < 0.001).

complications compared to the laparoscopic approach (12.8 vs 7.8%, p = 0.01). Conversely, patients undergoing the robotic-assisted approach had a lower rate of overall complications compared to the open approach (12.8 vs 18.5%, p = 0.02). After risk-adjustment, the robotic-assisted approach was linked to reduced odds of overall complications (AOR 0.63, 95% CI 0.44–0.92) when compared to the open approach, and increased odds of overall complications relative to the laparoscopic approach (AOR 1.86, 95% CI 1.16–3.00) (Table 4).

## Discussion

The optimal surgical technique for colostomy reversal remains controversial as both minimally invasive and open approaches present distinct implications for postoperative recovery, perioperative complications, and overall patient outcomes [17]. The present national analysis identified a notable increase in the utilization of minimally invasive colostomy reversal throughout the study period. Moreover, the minimally invasive approach was associated with significantly reduced odds of 30-day infectious, respiratory, and wound-related complications when compared to the open technique. Patients undergoing minimally invasive colostomy reversal also experienced notably shorter LOS, operative duration, and reduced rates of 30-day readmission. Given the scarce data regarding the optimal colostomy takedown approach in the current body of literature, our findings warrant further discussion.

The increasing adoption of minimally invasive colostomy reversal mirrors a broader trend toward minimally invasive techniques in colorectal surgery [7,18]. Congruent with our findings, Pei et al. have analyzed a national cohort of patients

**Table 1. Baseline characteristics of patients undergoing colostomy takedown from 2012–2022, stratified by surgical approach. Categorical data are expressed as % and continuous data are expressed as median [interquartile range, IQR].** *ASA*, American Society of Anesthesiologists; *BMI*, Body Mass Index; *COPD*, Chronic Obstructive Pulmonary Disease.

| | Minimally Invasive (n=6,180) | Open (n=13,983) | p-value |
|---|---|---|---|
| **Characteristics** | | | |
| Age (years, median, IQR) | 60 [49–69] | 60 [50–69] | 0.04 |
| Female (%) | 47.1 | 47.6 | 0.45 |
| ASA Class (%) | | | <0.001 |
| 1 | 2.0 | 2.1 | |
| 2 | 48.8 | 45.7 | |
| 3 | 46.9 | 49.2 | |
| 4 | 2.3 | 3.0 | |
| Race (%) | | | <0.001 |
| White | 78.7 | 74.8 | |
| Black | 8.2 | 10.0 | |
| Asian or Pacific Islanders | 2.4 | 1.7 | |
| Other/unknown | 10.7 | 13.5 | |
| Ethnicity (%) | | | <0.001 |
| Hispanic | 11.7 | 9.1 | |
| Functional Status (%) | | | 0.17 |
| Independent | 98.3 | 97.9 | |
| Partially Dependent | 1.6 | 2.0 | |
| Totally Dependent | 0.1 | 0.1 | |
| BMI, kg/m2 (%) | | | 0.39 |
| <18.5 | 1.7 | 2.0 | |
| 18.5-24.9 | 27.7 | 26.7 | |
| 25-29.9 | 33.6 | 33.4 | |
| 30-34.9 | 21.5 | 21.9 | |
| 35-39.9 | 9.6 | 9.6 | |
| >40 | 5.9 | 6.4 | |
| Comorbidities (%) | | | |
| COPD | 3.8 | 5.1 | <0.001 |
| Diabetes | 11.6 | 11.8 | 0.58 |
| Dyspnea | 4.1 | 5.0 | 0.06 |
| Hypertension | 44.6 | 46.8 | 0.01 |
| Smoker | 21.2 | 25.3 | <0.001 |
| Steroid use | 6.0 | 5.2 | 0.03 |
| Weight loss | 1.2 | 1.7 | 0.03 |

from 2005 to 2014 and found an average annual increase of 2.9% in the proportion of minimally invasive colostomy reversal procedures performed [7]. While this trend has continued beyond 2014, as demonstrated in the present study, a plateau is anticipated as certain patients remain unsuitable candidates for the minimally invasive approach. Namely, open colostomy reversal may be preferred in patients with multiple prior abdominal surgeries, clinically significant cardiovascular and pulmonary comorbidities, or a shortened rectal stump [19–24]. Nonetheless, recent advances in minimally invasive technology, increased surgeon proficiency, and growing recognition of the benefits of minimally invasive surgery

**Table 2. Unadjusted outcomes of colostomy takedown patients from 2012–2022, stratified by surgical approach.** IQR, interquartile range; LOS, Length of Stay (days). Outcomes were reported as proportions (%) for categorical data and as median [interquartile range] for continuous data.

| | Minimally Invasive (n = 6,180) | Open (n = 13,983) | p-value |
|---|---|---|---|
| **Primary Outcome** | | | |
| Overall Complications [a] | 13.2 | 23.1 | <0.001 |
| **Clinical Outcomes** | | | |
| Cardiac | 0.4 | 0.8 | <0.001 |
| Respiratory | 1.5 | 2.9 | <0.001 |
| Infectious | 8.6 | 15.9 | <0.001 |
| Wound | 0.6 | 1.7 | <0.001 |
| Renal | 0.5 | 1.0 | <0.001 |
| Thromboembolic | 0.4 | 0.9 | <0.001 |
| Transfusion | 2.1 | 4.6 | <0.001 |
| Reoperation | 4.0 | 5.3 | <0.001 |
| **Resource Utilization** | | | |
| Operative time (minutes) | 185 [133–254] | 196 [138–270] | <0.001 |
| LOS (days) | 4 [3–5 ] | 5 [4–7 ] | <0.001 |
| Nonhome Discharge | 3.1 | 5.4 | <0.001 |
| 30-Day Readmissions | 7.8 | 10.6 | <0.001 |

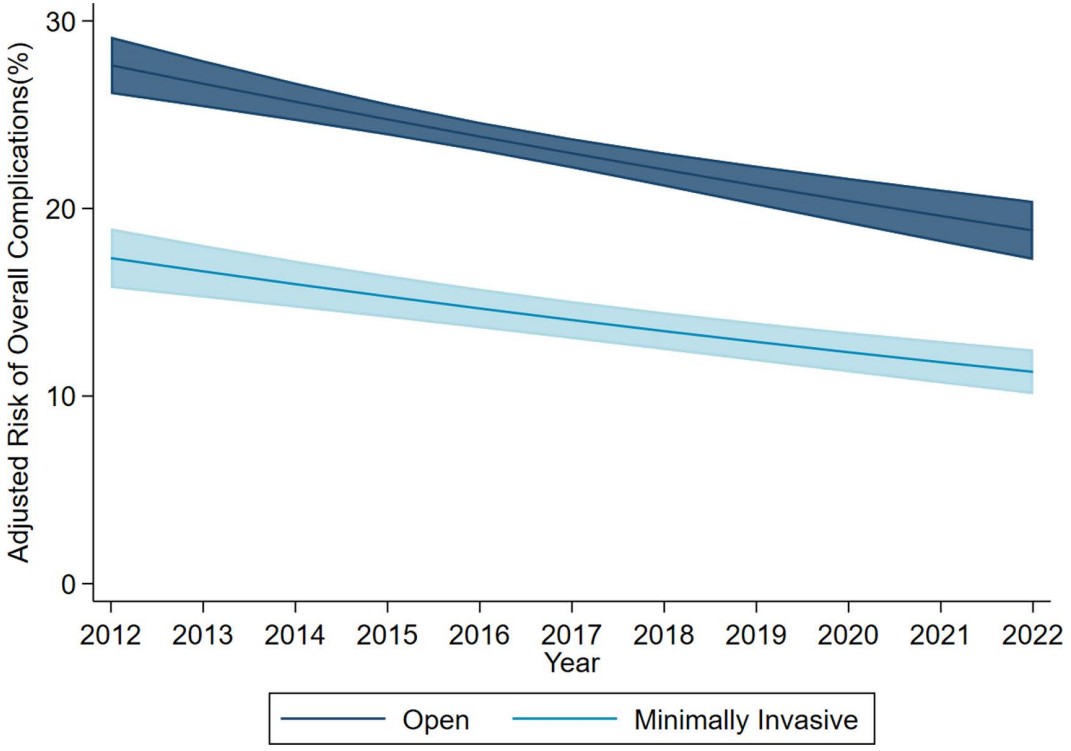

**Fig 3. Adjusted probability of overall complications among patients undergoing treatment for colostomy reversal, stratified by minimally invasive or open approach.** 95% confidence intervals are represented by shaded regions (2012–2022).

**Table 3. Adjusted outcomes of patients undergoing minimally invasive colostomy takedown from 2012–2022 (Reference: Open approach).** IQR, interquartile range; LOS, Length of Stay (days). Adjusted outcomes are reported as Odds Ratio (AOR) with 95% confidence intervals (95%CI) or β coefficient (β), as appropriate.

| | AOR/β with 95% CI | p-value |
|---|---|---|
| **Primary Outcome** | | |
| Overall Complications[a] | 0.56 [0.51-0.62] | <0.001 |
| **Complications** | | |
| Cardiac | 0.55 [0.34-0.89] | 0.02 |
| Respiratory | 0.61 [0.47-0.79] | <0.001 |
| Infectious | 0.54 [0.48-0.61] | <0.001 |
| Wound | 0.39 [0.26-0.57] | <0.001 |
| Renal | 0.60 [0.40-0.89] | 0.01 |
| Thromboembolic | 0.48 [0.31-0.73] | 0.001 |
| Transfusion | 0.54 [0.44-0.66] | <0.001 |
| Reoperation | 0.80 [0.67-0.95] | 0.01 |
| **Resource Utilization** | | |
| Operative time (minutes) | −16.9 [−20.2, −13.6] | <0.001 |
| LOS (days) | −1.70 [−1.84, −1.56] | <0.001 |
| Nonhome Discharge | 0.59 [0.48-0.71] | <0.001 |
| 30-Day Readmissions | 0.75 [0.67-0.85] | <0.001 |

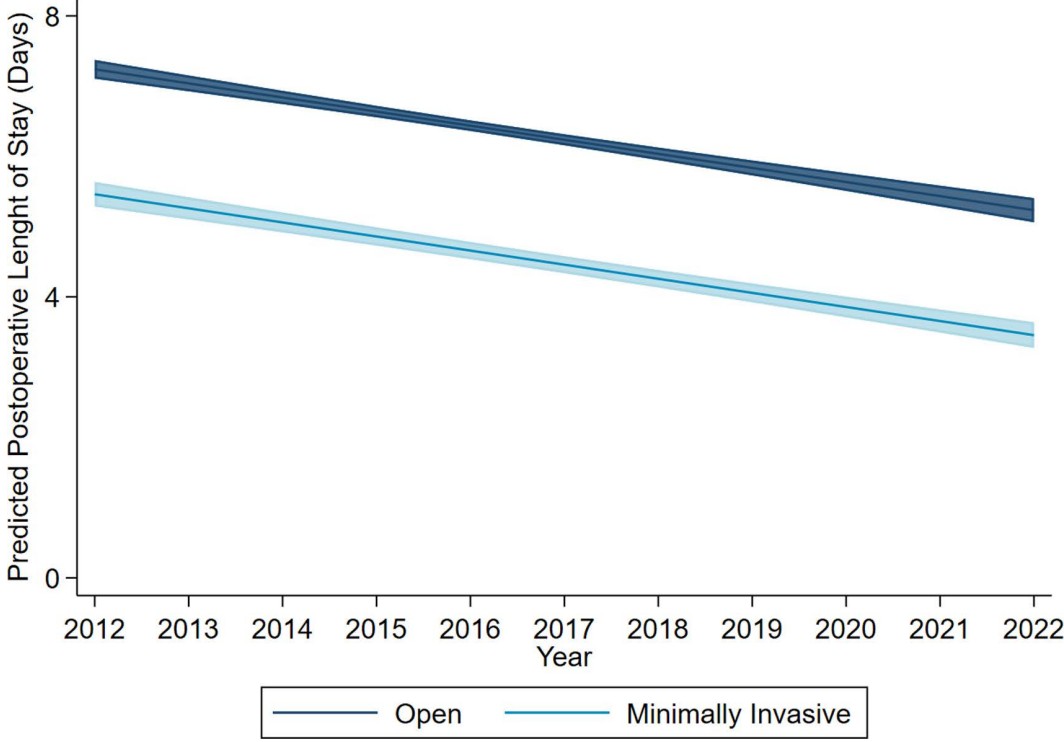

**Fig 4. Predicted postoperative length of stay among patients undergoing colostomy reversal, stratified by minimally invasive or open approach.** 95% confidence intervals are represented by shaded regions (2012–2022).

**Table 4. Adjusted outcomes of patients undergoing robotic-assisted colostomy takedown in reference to both open and laparoscopic approaches (2022). IQR, interquartile range; LOS, Length of Stay (days). Adjusted outcomes are reported as Odds Ratio (AOR) with 95% confidence intervals (95%CI) or β coefficient (β), as appropriate.**

| | Robotic-Assisted approach (Reference: Laparoscopic) | | Robotic-Assisted approach (Reference: Open) | |
|---|---|---|---|---|
| | AOR/β with 95% CI | p-value | AOR/β with 95% CI | p-value |
| **Primary Outcome** | | | | |
| Overall Complications | 1.86 (1.16-3.00) | 0.01 | 0.63 (0.44–0.92) | 0.02 |
| **Complications** | | | | |
| Cardiac | 1.26 (0.84–1.89) | 0.27 | 0.29 (0.36–2.31) | 0.24 |
| Respiratory | 2.20 (0.47-10.3) | 0.31 | 0.58 (0.22–1.53) | 0.27 |
| Infectious | 2.07 (1.23–3.49) | 0.01 | 0.69 (0.46–1.04) | 0.07 |
| Wound | 0.58 (0.05-7.08) | 0.67 | 0.27 (0.03–2.07) | 0.21 |
| Renal | 0.90 (0.41-2.02) | 0.81 | 0.56 (0.29–1.09) | 0.09 |
| Thromboembolic | 0.98 (0.85-1.14) | 0.80 | 1.01 (0.96–1.06) | 0.84 |
| Transfusion | 0.52 (0.20-1.35) | 0.18 | 0.44 (0.19–1.06) | 0.07 |
| Reoperation | 1.97 (0.87-4.46) | 0.11 | 0.69 (0.37-1.28) | 0.24 |
| **Resource Utilization** | | | | |
| Operative time (minutes) | 36.3 (22.69-49.99) | <0.001 | 18.7 (5.08-32.3) | 0.01 |
| LOS (days) | 0.11 (−0.29, 0.52) | 0.58 | −1.46 (−1.93, −0.98) | <0.001 |
| Nonhome Discharge | 2.75 (1.10-6.87) | 0.03 | 1.35 (0.72-2.54) | 0.35 |
| 30-Day Readmissions | 1.11(0.66-1.89) | 0.69 | 0.69 (0.43-1.08) | 0.11 |

likely underpin this shift toward minimally invasive reversal [25,10]. Patient preferences and the accumulation of supportive evidence may have further encouraged the uptake of minimally invasive techniques, resulting in more widespread availability, improved resource utilization, and potentially lower perioperative complication rates [10,26], Furthermore, the continued expansion of minimally invasive training programs may enhance the accessibility of minimally invasive options for a broader patient population, thereby optimizing financial and clinical outcomes of colostomy takedown [27].

Corroborating prior work, our study found minimally invasive colostomy reversal to be associated with a significant reduction in respiratory, infectious, and wound complications [28]. Namely, reducing incision size minimizes exposure of the surgical field to external contaminants, thereby lowering the risk of surgical site infections, which are significant contributors to postoperative complications [17,28]. Minimally invasive techniques offer advantages over open approaches, including enhanced visualization of the splenic flexure and the ability to minimize the extent of the midline incision, thereby potentially reducing postoperative morbidity [17,29]. Adequate mobilization of the splenic flexure is critical to prevent undue tension at the anastomotic site, which can lead to complications such as anastomotic dehiscence or stricture formation [17,29]. Given these benefits, it is essential to further examine how minimally invasive techniques compare across different patient populations and surgical settings, particularly in terms of long-term functional outcomes and cost-effectiveness.

Furthermore, in contrast to the prolonged operative times often reported in minimally invasive colectomy for conditions such as diverticulitis, colon cancer, and ischemic colitis, our analysis identified decreased operative durations with minimally invasive colostomy reversal [18,30,31]. While two large meta-analyses previously reported comparable operative times between minimally invasive and open approaches, [8,32] more recent investigations, similar to the present study, have demonstrated shorter operative durations in the minimally invasive cohort [7,10,28]. One plausible explanation is that minimally invasive colostomy reversal generally requires primarily the creation of an anastomosis, which is inherently less technically demanding than more extensive minimally invasive procedures [30,31,33]. The smaller incisions in the minimally invasive approach also eliminate the time required to open and close the abdomen, potentially explaining

the shorter operative times observed. Moreover, cases with severe abdominal adhesions may often require the open approach, leading to increased operative time [10].

Beyond reduced operative times, minimally invasive colostomy reversal was associated with shorter postoperative length of stay as well as reduced rates of 30-day readmission. Similar benefits have been well-documented in the broader field of gastrointestinal surgery, where patients undergoing minimally invasive procedures often experience faster recoveries, quicker return to normal activities, and fewer readmissions [17,28,29]. These benefits are also observed in patients undergoing colostomy reversal and may be attributed to several factors, including reduced postoperative pain, expedited recovery of bowel function, and a shorter time to resumption of diet [10,34–36]. Notably, we showed that postoperative LOS has continued to decrease over the study period, with the minimally invasive cohort consistently experiencing reduced LOS compared to open. This trend may be attributable to the growing adoption of Enhanced Recovery After Surgery (ERAS) protocols for both open and minimally invasive procedures [37,38]. Given that an estimated 750,000–1 million individuals in the United States live with an ostomy, our findings suggest a substantial patient population that may benefit from minimally invasive colostomy reversal [39]. By shortening hospital stays and reducing readmission rates, minimally invasive approaches can potentially free hospital resources for more patients, ultimately easing the healthcare system's burden and improving overall patient well-being [40–42].

In the robotic-assisted subgroup analysis, the robotic-assisted approach was associated with reduced overall complications compared to the open approach. Notably, this differs from recent literature that relies on insurance diagnostic codes for stratification, as our analysis utilized a validated, peer-controlled database with standardized data collection by dedicated reviewers and robust interrater validation [43,44]. The robotic-assisted approach offers significant advantages, including enhanced visualization and precision during dissection, especially in confined pelvic spaces and around the colostomy site [45]. These capabilities make it well-suited for complex cases, such as extensive adhesiolysis [45]. Nevertheless, the robotic-assisted approach exhibited higher overall complications relative to the laparoscopic approach. This discrepancy is likely multifactorial, in part due to differences in surgeon experience. While laparoscopic surgery is well-established, many surgeons may still be developing proficiency in robotic-assisted techniques [44,46]. Colostomy reversals inherently require exploration of the previously operated abdomen and pelvis, often complicated by significant inflammatory changes from the initial surgery. Laparoscopic techniques facilitate easy and rapid adjustments in forceps, trocar placement, and patient positioning, making them particularly advantageous for managing the complexities of re-operative procedures in these regions [44,47]. Nevertheless, a recent study by Katsura et al. found that the proportion of robotic-assisted colostomy reversals increased significantly between 2015 and 2020 [44]. It is reasonable to anticipate that as adoption and training in robotic-assisted approaches continue to grow in the coming years, the associated complication rates will subsequently decrease.

This study has several important limitations including those inherent to its retrospective design. NSQIP-participating hospitals are known to be higher-volume teaching institutions and do not necessarily represent all hospital types in the United States [48]. Our study was unable to delineate overall trends between laparoscopic and robotic-assisted colostomy reversals, as the ACS NSQIP began coding for robotic-assisted procedures only in 2022. Consequently, a significant portion of the adoption of minimally invasive surgery may be attributable to the increased utilization of robotic-assisted techniques [44]. Moreover, it is possible that prior to 2022, a portion of robotic surgeries were classified under laparoscopic codes. Additionally, the modality of the patients' initial ostomy placement, whether minimally invasive or open, remains unidentified. NSQIP lacked adequate data on the conversion of laparoscopic colostomy reversals to open procedures prior to the year 2022 [12]. Moreover, the retrospective design of our study introduces the potential for selection bias, as surgeons may be more inclined to pursue laparoscopic colostomy reversal in patients with less hostile intra-abdominal conditions. Conversely, open approaches may be preferentially selected for more complex cases, such as those with a history of perforated diverticulitis, thereby potentially confounding comparisons of postoperative outcomes. Lastly, outcomes in NSQIP are only recorded for 30 days postoperatively, preventing the further analysis of adverse events beyond

this period. Future studies may seek to evaluate longer-term outcomes as well as patient-reported outcomes measures after colostomy, whether performed by minimally invasive or open approach. Nonetheless, the present study utilized robust statistical methodology to mitigate the impact of these limitations.

The utilization of minimally invasive colostomy reversal has significantly increased over recent years and is associated with lower rates of postoperative complications, shorter hospital stays, and reduced readmission rates compared to the open approach. These findings suggest that minimally invasive colostomy reversal offers superior short-term outcomes and should be considered as a preferred surgical option when feasible. Efforts to increase surgeon proficiency in minimally invasive techniques and to address barriers to procedural adoption may further improve patient outcomes. However, future prospective studies to evaluate long-term outcomes associated with minimally invasive versus open colostomy reversal are warranted.

## Supporting information

**S1 Table. Variables Selected via Least Absolute Shrinkage and Selection Operator (LASSO) for Multivariable Modeling.** *Abbreviations:* **ASA, American Society of Anesthesiologists; BMI, Body Mass Index; CHF, Congestive Heart Failure; COPD, Chronic Obstructive Pulmonary Disease; HTN, Hypertension.**
(DOCX)

## Author contributions

**Conceptualization:** Kevin Tabibian, Peyman Benharash.

**Data curation:** Kevin Tabibian, Nam Yong Cho, Oh Jin Kwon, Esteban Aguayo, Dariush Yalzadeh, Arjun Chaturvedi.

**Formal analysis:** Kevin Tabibian, Nam Yong Cho, Oh Jin Kwon, Esteban Aguayo, Hanjoo Lee.

**Investigation:** Kevin Tabibian, Oh Jin Kwon, Hanjoo Lee.

**Methodology:** Kevin Tabibian, Esteban Aguayo, Ayesha P Ng.

**Software:** Kevin Tabibian, Nam Yong Cho.

**Supervision:** Nam Yong Cho, Esteban Aguayo, Peyman Benharash, Hanjoo Lee.

**Validation:** Kevin Tabibian, Oh Jin Kwon.

**Visualization:** Peyman Benharash.

**Writing – original draft:** Kevin Tabibian, Courtney Obasohan, Hanjoo Lee.

**Writing – review & editing:** Nam Yong Cho, Oh Jin Kwon, Esteban Aguayo, Ayesha P Ng, Arjun Chaturvedi, Peyman Benharash, Hanjoo Lee.

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
