## [Decision Letter · Decision Letter 0]

Dear Dr. Lee,

Thank you for submitting your manuscript to PLOS ONE. After careful consideration, we feel that it has merit but does not fully meet PLOS ONE’s publication criteria as it currently stands. Therefore, we invite you to submit a revised version of the manuscript that addresses the points raised during the review process.

We look forward to receiving your revised manuscript.

Kind regards,

Stanisław Jacek Wroński, M.D., Ph.D, FEBU

Academic Editor

PLOS ONE

Additional Editor Comments:

Dear Authors,

thank you for an interesting article presenting a retrospective study. However, the reviewers feel that some points need clarification and/or improvement. Please read the comments below. We hope you will find them helpful in rewriting the article to make it fully suitable for publication in PLOS ONE:

Reviewer 1:

1. This is a well-powered retrospective study, so it was expected to see so

many significant results. I commend the authors, though, for taking a critical approach at reviewing these findings and honing in on areas of larger margins of significance.

2. It is refreshing to see correct applications for LASSO ridge regression and multivariable logistic and linear regression. However, the use of these models and the results reported suggest adjustment of some kind. Considering the exploratory and validating nature of this study, a supplementary table or some additional details on what variables were retained during the LASSO process would have been beneficial for clarity and replicability.

Reviewer 2:

This is not a randomized study and it seems likely that selection bias had an impact ie. surgeons would be more likely to attempt laparoscopic ostomy takedown if they thought they would encounter a less hostile abdomen whereas a patient whose first operation was very difficult eg. perforated diverticulitis might favor a open approach. This makes the results difficult to interpret.

2. Although the CPT codes used included both ileostomy and colostomy, the authors excluded the ileostomy cases using ICD 9/10

codes. How accurate do the authors think this approach was?

3. The authors excluded patients that were attempted laparoscopically but required conversion to open. Shouldn't this analysis be intent to treat or at least have included this group as a separate analysis? How large was this group?

Reviewer 3:

1. Postoperative complications were chosen as the study’s primary endpoints. Although the study has the inherent limitations of a retrospective design, a rigorous statistical analysis was conducted to balance those limitations. Their results show promising developments in favour of minimally invasive surgery regarding a procedure where an open approach was preferred.

with compliments

Stanisław Wroński

Academic Editor

Reviewers' comments:

Reviewer's Responses to Questions

**Comments to the Author**

1. Is the manuscript technically sound, and do the data support the conclusions?

Reviewer #1: Yes

Reviewer #2: Partly

Reviewer #3: Yes

2. Has the statistical analysis been performed appropriately and rigorously?

Reviewer #1: Yes

Reviewer #2: Yes

Reviewer #3: Yes

3. Have the authors made all data underlying the findings in their manuscript fully available?

Reviewer #1: Yes

Reviewer #2: Yes

Reviewer #3: Yes

4. Is the manuscript presented in an intelligible fashion and written in standard English?

Reviewer #1: Yes

Reviewer #2: Yes

Reviewer #3: Yes

Reviewer #1: Thank you for the opportunity to review this study. Validating current surgical trends within colorectal care is an important step in understanding and devising gold standards of practice. This is a well-powered retrospective study, so it was expected to see so many significant results. I commend the authors, though, for taking a critical approach at reviewing these findings and honing in on areas of larger margins of significance.

It is refreshing to see correct applications for LASSO ridge regression and multivariable logistic and linear regression. However, the use of these models and the results reported suggest adjustment of some kind. Considering the exploratory and validating nature of this study, a supplementary table or some additional details on what variables were retained during the LASSO process would have been beneficial for clarity and replicability.

Nonetheless, this does not completely detract from the study, which is worthwhile for validating current changes in surgical techniques and the risk of complications.

Reviewer #2: Suggestions to improve the manuscript:

1. This is not a randomized study and it seems likely that selection bias had an impact ie. surgeons would be more likely to attempt laparoscopic ostomy takedown if they thought they would encounter a less hostile abdomen whereas a patient whose first operation was very difficult eg. perforated diverticulitis might favor a open approach. This makes the results difficult to interpret.

2. Although the CPT codes used included both ileostomy and colostomy, the authors excluded the ileostomy cases using ICD 9/10 codes. How accurate do the authors think this approach was?

3. The authors excluded patients that were attempted laparoscopically but required conversion to open. Shouldn't this analysis be intent to treat or at least have included this group as a separate analysis? How large was this group?

Reviewer #3: Thank you for the opportunity to review this manuscript.

The authors have analysed a large national retrospective cohort comparing minimally invasive vs. open colostomy reversal. Postoperative complications were chosen as the study’s primary endpoints. Although the study has the inherent limitations of a retrospective design, a rigorous statistical analysis was conducted to balance those limitations. Their results show promising developments in favour of minimally invasive surgery regarding a procedure where an open approach was preferred. This manuscript should be accepted, as it encourages colorectal surgeons to further the trend toward minimally invasive surgery in this field.

**Do you want your identity to be public for this peer review?** For information about this choice, including consent withdrawal, please see our Privacy Policy

Reviewer #1: No

Reviewer #2: No

Reviewer #3: No

---

## [Author Response · Author response to Decision Letter 1]

26 May 2025

May 26, 2025

Revision Note for Manuscript Entitled “National Trends in Utilization and Outcomes of Elective Open and Minimally Invasive Colostomy Reversal: A NSQIP Analysis”

Reviewer #1:

1. This is a well-powered retrospective study, so it was expected to see so

many significant results. I commend the authors, though, for taking a critical approach at reviewing these findings and honing in on areas of larger margins of significance.

We thank the reviewer for their positive appraisal.

2. It is refreshing to see correct applications for LASSO ridge regression and multivariable logistic and linear regression. However, the use of these models and the results reported suggest adjustment of some kind. Considering the exploratory and validating nature of this study, a supplementary table or some additional details on what variables were retained during the LASSO process would have been beneficial for clarity and replicability.

We appreciate the reviewer’s perceptive comment. In agreement, we have added a supplementary table listing the specific variables retained during the LASSO process to enhance transparency and reproducibility.

Reviewer #2:

1. This is not a randomized study and it seems likely that selection bias had an impact ie. surgeons would be more likely to attempt laparoscopic ostomy takedown if they thought they would encounter a less hostile abdomen whereas a patient whose first operation was very difficult eg. perforated diverticulitis might favor a open approach. This makes the results difficult to interpret

We thank the reviewer for highlighting this critical point. We acknowledge that the retrospective nature of our study introduces inherent selection bias, as surgeons may preferentially choose laparoscopic ostomy takedown for patients with less hostile abdomens and opt for open approaches in more complex cases. This limitation, also noted in prior studies [1,2], may influence the interpretation of our results. To address this, we have revised the limitations section of our manuscript.

Page 17, Lines 322–327

“Moreover, the retrospective design of our study introduces the potential for selection bias, as surgeons may be more inclined to pursue laparoscopic colostomy reversal in patients with less hostile intra-abdominal conditions. Conversely, open approaches may be preferentially selected for more complex cases, such as those with a history of perforated diverticulitis, thereby potentially confounding comparisons of postoperative outcomes”

[1] Pei KY, Davis KA, Zhang Y. Assessing trends in laparoscopic colostomy reversal and evaluating outcomes when compared to open procedures. Surgical Endoscopy. 2018 Feb;32:695-701.

[2] Katsura M, Ashbrook M, Ikenoue T, Takahashi K, Ito MA, Martin MJ, Inaba K, Matsushima K. Surgical trends and outcomes of open, laparoscopic, and robotic colostomy reversal for benign disease. Surgery. 2024 Nov 1;176(5):1366-73.

2. Although the CPT codes used included both ileostomy and colostomy, the authors excluded the ileostomy cases using ICD 9/10 codes. How accurate do the authors think this approach was?

We thank the reviewer for this insightful observation. Earlier NSQIP analyses of Hartmann’s reversal used the same CPT framework but did not exclude ileostomy cases, leaving room for misclassification bias [1,2]. Our methodology improves upon these prior approaches and aligns with the strategy more recently employed by Nasseri et al., who utilized the same CPT codes and similarly excluded ileostomy cases using validated ICD codes [3]. While no administrative algorithm is flawless, we are confident that this two-step strategy yields the most accurate colostomy-reversal cohort currently attainable within NSQIP.

[1] Kooragayala K, Lou J, Butchy V, Balakrishnan A, Sandilos G, Kwiatt M, Giugliano D, McClane S. Impact of frailty on patient outcomes after Hartmann’s reversal: A NSQIP analysis. The American Surgeon™. 2023 Dec;89(12):5459-65.

[2] Pei KY, Davis KA, Zhang Y. Assessing trends in laparoscopic colostomy reversal and evaluating outcomes when compared to open procedures. Surgical Endoscopy. 2018 Feb;32:695-701.

[3] Nasseri Y, Liu A, Kasheri E, Oka K, Langenfeld S, Smiley A, Cohen J, Ellenhorn J, Barnajian M. Hartmann's reversal is associated with worse outcomes compared to elective left colectomy: A NSQIP analysis of 36,794 cases. The American Journal of Surgery. 2022 Dec 1;224(6):1351-5.

3. The authors excluded patients that were attempted laparoscopically but required conversion to open. Shouldn't this analysis be intent to treat or at least have included this group as a separate analysis? How large was this group?

We thank the reviewer for their astute observation. The NSQIP database did not begin recording the “conversion-to-open” variable until the 2022 data release; consequently, no conversion information is available for 2012-2021. In the 2022 file we identified only 190 conversion cases, an insufficient sample size for a powered subgroup analysis.

We agree with the reviewer: the primary analysis (2012-2022) was conducted on an intent-to-treat basis, classifying procedures by their initial approach. To address the potential impact of conversions, we performed a 2022 sensitivity analysis which excluded converted cases. Results were substantively unchanged, supporting the robustness of our main findings.

Reviewer #3:

1. Postoperative complications were chosen as the study’s primary endpoints. Although the study has the inherent limitations of a retrospective design, a rigorous statistical analysis was conducted to balance those limitations. Their results show promising developments in favour of minimally invasive surgery regarding a procedure where an open approach was preferred.

We greatly appreciate the reviewer’s positive appraisal of our manuscript.

---

## [Decision Letter · Decision Letter 1]

National Trends in Utilization and Outcomes of Elective Open and Minimally Invasive Colostomy Reversal: A NSQIP Analysis

PONE-D-25-12577R1

Dear Dr. Hanjoo Lee

We’re pleased to inform you that your manuscript has been judged scientifically suitable for publication and will be formally accepted for publication once it meets all outstanding technical requirements.

Kind regards,

Stanisław Jacek Wroński, M.D., Ph.D, FEBU

Academic Editor

PLOS ONE

Additional Editor Comments (optional):

Dear Authors,

after careful consideration of the reviewers' opinions on the original version of the article and the additional opinion of one reviewer on the second, revised version, I conclude that the submitted paper entitled “National Trends in Utilization and Outcomes of Elective Open and Minimally Invasive Colostomy Reversal: A NSQIP Analysis” (PONE-D-25-12577R1) meets the requirements for publication in PLOS ONE. All comments have been addressed

With compliments

Stanisław Wroński

Academic Editor

Reviewers' comments:

Reviewer's Responses to Questions

**Comments to the Author**

Reviewer #2: All comments have been addressed

2. Is the manuscript technically sound, and do the data support the conclusions?

Reviewer #2: (No Response)

3. Has the statistical analysis been performed appropriately and rigorously?

Reviewer #2: (No Response)

4. Have the authors made all data underlying the findings in their manuscript fully available?

Reviewer #2: (No Response)

5. Is the manuscript presented in an intelligible fashion and written in standard English?

Reviewer #2: (No Response)

Reviewer #2: (No Response)

**Do you want your identity to be public for this peer review?** For information about this choice, including consent withdrawal, please see our Privacy Policy

Reviewer #2: No

---

## [Editor Report · Acceptance letter]

PONE-D-25-12577R1

PLOS ONE

Dear Dr. Lee,

I'm pleased to inform you that your manuscript has been deemed suitable for publication in PLOS ONE. Congratulations! Your manuscript is now being handed over to our production team.

Kind regards,

on behalf of

Dr. Stanisław Jacek Wroński

Academic Editor

PLOS ONE